# THE JOURNEY MATTERS:
# AVERAGE PARAMETER COUNT OVER PRE-TRAINING UNIFIES SPARSE AND DENSE SCALING LAWS

**Tian Jin**[1]* **Ahmed Imtiaz Humayun**[2,3] **Utku Evci**[4] **Suvinay Subramanian**[5]
**Amir Yazdanbakhsh**[4] **Dan Alistarh**[6] **Gintare Karolina Dziugaite**[4]
[1]MIT CSAIL  [2]Rice University  [3]Google Research  [4]Google DeepMind  [5]Google  [6]IST Austria

## ABSTRACT

Pruning eliminates unnecessary parameters in neural networks; it offers a promising solution to the growing computational demands of large language models (LLMs). While many focus on post-training pruning, sparse pre-training–which combines pruning and pre-training into a single phase–provides a simpler alternative. In this work, we present the first systematic exploration of optimal sparse pre-training configurations for LLMs through an examination of 80 unique pruning schedules across different sparsity levels and training durations. We find that initiating pruning at 25% of total training compute and concluding at 75% achieves near-optimal final evaluation loss. These findings provide valuable insights for efficient and effective sparse pre-training of LLMs. Furthermore, we propose a new scaling law that modifies the Chinchilla scaling law to use the *average* parameter count over pre-training. Through empirical and theoretical validation, we demonstrate that this modified scaling law accurately models evaluation loss for both sparsely and densely pre-trained LLMs, unifying scaling laws across pre-training paradigms. Our findings indicate that while sparse pre-training achieves the same final model quality as dense pre-training for equivalent compute budgets, it provides substantial benefits through reduced model size, enabling significant potential computational savings during inference.

## 1 INTRODUCTION

As language models grow in size and train on more data, they consistently demonstrate improved performance (Brown et al., 2020; Kaplan et al., 2020; Hoffmann et al., 2024; Nakkiran et al., 2020). However, their enormous size poses an increasingly pressing challenge to their efficient deployment and equitable access. *Sparse pre-training* integrates neural network pruning (Han et al., 2015; LeCun et al., 1989; Hassibi et al., 1993; He et al., 2017) into pre-training and offers a promising solution to these challenges by activating only a subset of parameters during both training and inference, reducing computational costs; it gained prominence when Lottery Ticket Hypothesis (Frankle & Carbin, 2019) presented compelling evidence for its feasibility. Subsequent work introduced a series of sparse pre-training algorithms (Evci et al., 2020; Peste et al., 2021; Kuznedelev et al., 2024).

While a growing body of research investigates pruning large language models (LLMs) post-training (Sun et al., 2024; Frantar & Alistarh, 2023; Xia et al., 2023), our work focuses on sparse pre-training. The combined challenges of LLM pre-training costs and pruning algorithm design present substantial obstacle to this direction of research. For example, Lottery Ticket Hypothesis (Frankle & Carbin, 2019) identifies trainable sparse sub-networks using iterative pruning and retraining, a process that becomes prohibitively expensive at the scale of contemporary LLMs. This enormous expense limits investigation to smaller-scale models, leaving the optimal strategies for sparse pre-training of LLMs largely unknown. *Scaling laws* – which predict how language modeling loss varies with model and data size – can help us extend insights from small-scale experiments to large models.

This leads to a critical question: how does sparsity change the scaling laws, which inform practical LLM training design (Sardana et al., 2024; Grattafiori et al., 2024)?

---

*Correspondence to: `tianjin@csail.mit.edu`, `gkdz@google.com`

Previous work on sparse pre-training by Frantar et al. (2024) introduced a new sparsity-aware scaling law that departed from the established scaling laws for dense models. The modified laws included extra terms dependent on the final sparsity, aiming to capture the sparsity-specific scaling effects.

**Our approach.** Instead, we revisit the original dense scaling laws and explore how to adjust the parameter count term for sparse pre-training, where it gradually decreases over the course of pre-training. Our experiments show that dense scaling laws effectively model sparse pre-training when using the *average parameter count* – the mean of per-step parameter count over pre-training. Our findings suggest that the core principles of dense scaling laws remain applicable in the sparse pre-training regime, with the key adjustment being a more nuanced consideration of model size.

**Optimal sparse pre-training configuration.** To validate this approach, we evaluate over 80 combinations of sparse pre-training schedules, sparsity levels, and training durations. Our work provides the first systematic analysis of sparse pre-training configurations across design dimensions critical to sparse pre-training. Our systematic evaluation uncovers the optimal configurations and offers new insights into the dynamics of sparse pre-training.

Our results show that, for a fixed compute budget, initiating pruning at 25% of total training compute and concluding at 75% achieves near-optimal final evaluation loss across various training durations and sparsity levels. We find that sparse pre-training performs best using the same optimal learning rates and batch sizes as the initial dense model under equivalent compute budgets. We also provide additional analysis into failure modes that occur with non-optimal sparse pre-training configurations, highlighting the importance of a properly tuned configuration. Collectively, our analysis presents practical prescriptions that ease the transition from dense to sparse pre-training configurations.

**Scaling analysis.** We extend the Chinchilla scaling law to model sparsely pre-trained LLMs by using the average parameter count over pre-training. We only count active parameters, which receive gradient updates; they become inactive once pruned. This adaptation allows us to predict evaluation loss across a range of model sizes, sparsity levels, and training durations. Using empirically verified assumptions about sparse pre-training dynamics, we provide theoretical justification for why average parameter count accurately models the evaluation loss of sparsely pre-trained LLMs.

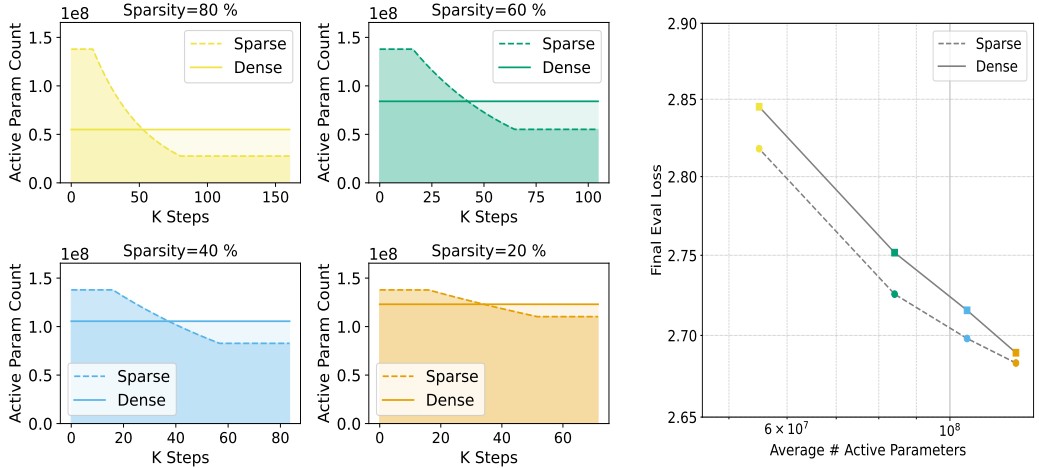

Figure 1: We show the predictive power of average active parameters by creating two families of models. The first is sparse, starting from a dense model with 138 million prunable parameters in the linear layers and targeting final sparsity levels of 20%, 40%, 60%, and 80%. The second is dense, created by adjusting the hidden dimension to match the average number of active parameters throughout sparse-pre-training for each sparse models. In the left plot, we represent sparse models with dashed lines and dense models with solid lines. Each sparse-dense pair, with matching average active parameters, is shown in the same subfigure. Each pairs of model shares the same total training compute. In the right plot, despite differences in pre-training techniques, sparse and dense models with matching average active parameters (indicated by matching colors) achieve similar final loss.

**Average parameter count.** To demonstrate how average parameter count predicts sparse pre-training loss, we pre-train and compare four pairs of sparse and dense models. In each pair, we match their average parameter count[1] and training token count. As shown in Figure 1, while each sparse-dense pair differs in parameter count during pre-training (left), they achieve similar final loss (right). This comparison demonstrates that average parameter count predicts final evaluation loss equally well for both sparse and dense models. In Appendix C, we replicate this result on a larger scale using models exceeding 1B parameters, and in Appendix D, we validate that sparse and dense pairs also match in downstream task evaluations.

**Contributions.** We make the following contributions in our work:

1. We unify sparse and dense scaling laws by modifying the parameter count term in the Chinchilla model and extending it to model sparse pre-training. This modified scaling law accurately models evaluation loss across model sizes, sparsity levels, and training durations.

2. We present a theoretical analysis, based on empirically validated assumptions about sparse pre-training dynamics, that justifies using average parameter count to model sparse pre-training loss.

3. We search over 80 sparse pre-training configurations and present a simple prescription that achieves optimal or near-optimal loss across different sparsity levels and training duration.

4. We present an analysis over the failure modes when sparse pre-training configurations deviate from the said prescription, highlighting their practical importance.

**Implications.** Our work enables practitioners to extend the familiar dense scaling principles to sparse pre-training. By providing optimal sparse pre-training configurations and a unified scaling law, our work facilitates the transition from dense to sparse pre-training, promoting the development of more computationally efficient large language models (LLMs).

## 2 RELATED WORK

**Neural scaling laws.** Neural scaling laws provide a framework for understanding how neural networks' performance scales with parameters, data, and compute (Banko & Brill, 2001; Goodman, 2001; Ghorbani et al., 2022; Kaplan et al., 2020; Hoffmann et al., 2024; Bansal et al., 2022; Gordon et al., 2021). Kaplan et al. (2020) showed that for modern transformer-based language models, model loss decreases predictably with increasing model size, dataset size, and compute, following a power-law relationship. Hoffmann et al. (2024) later refined these insights by optimizing hyperparameter configurations, such as learning rate schedules, and proposed a new scaling law that emphasizes scaling training data more aggressively than Kaplan et al. (2020)'s original recommendations. Most relevant to our work are Rosenfeld et al. (2021) and Frantar et al. (2024). Rosenfeld et al. (2021) focused on small-scale CNNs for image classification, while Frantar et al. (2024) focused on transformer-based vision and language models; both modeled their respective performance as a function of model size and pruning configurations. Our work is different in that we unify the functional forms of scaling laws for both dense and sparse pre-training. This unification is partly enabled by our novel exploration of optimal hyperparameter configurations for sparse pre-training.

Additionally, our study is the largest-scale investigation of sparsely pretrained LLMs to date, with our largest model using over 5 times the compute of the largest model examined in prior work (Frantar et al., 2024). The largest model we investigate requires $4.5 \times 10^{20}$ FLOPs training compute.

**Pruning.** Sparse pre-training involves pruning parameters in an LLM during the pre-training process. While there are many pruning algorithms available (LeCun et al., 1989; Hassibi et al., 1993; Han et al., 2015; He et al., 2017; Frankle & Carbin, 2019; Renda et al., 2020; Peste et al., 2021; Evci et al., 2020), we focus on a simple class of algorithms known as iterative magnitude pruning (Zhu & Gupta, 2017; Frankle & Carbin, 2019; Renda et al., 2020). Since sparse pre-training results in a sparse model at the end of training, our method can also be viewed as a pruning algorithm. In contrast to prevailing approaches that train a large dense model and then prune it while preserving accuracy (Frantar & Alistarh, 2023; Sun et al., 2024), our analysis shows that, within the sparsity

---

[1]We adjust the hidden dimension of dense models to match the sparse models, but since it must be divisible by the number of attention heads and chips, we create two dense models that best approximate the target parameter count and linearly interpolate between them.

levels we examined, it is possible to directly train a sparse model that achieves the same final loss using the same compute budget as training the large dense model. This simplifies the model development process by eliminating the need for pruning as a post-training step.

**Dynamic parameter schedule.** While practitioners typically pre-train LLMs with a fixed number of active parameters throughout the training process (Groeneveld et al., 2024; Shoeybi et al., 2020), a growing line of work explores varying this number during pre-training to improve computational efficiency (Yao et al., 2024; Panigrahi et al., 2024; Yano et al., 2024). This line of work often focuses on gradually increasing the number of parameters during training. Yao et al. (2024) proposed progressive growth during pre-training using a multi-stage, multi-axis growth schedule. Panigrahi et al. (2024) introduced layer dropout, progressively reducing the number of dropped layers during training. Yano et al. (2024) developed STEP, which begins pre-training with a small model and gradually increases its size in stages. Our work may be viewed as proposing another dynamic parameter schedule. However, it differs by focusing on compute-optimal strategies that gradually *reduce* model size, optimizing for inference efficiency. Since the final model is smaller, our approach leads to more efficient inference compared to methods that progressively increase model size.

## 3 PRELIMINARIES

**Algorithm.** We adopt the iterative magnitude pruning (IMP) algorithm for pre-training LLMs (Zhu & Gupta, 2017; Renda et al., 2020; Frankle & Carbin, 2019; Samar, 2022; Frantar et al., 2024). We score each parameter's importance based on its magnitude. At each pruning iteration, we rank all model parameters globally and prune those with the lowest magnitudes. No structural constraints are imposed on the sparsity pattern. Our sparse pre-training algorithm consists of three phases:

1. *Dense training phase*: We train the dense starting model with all its parameters for $N_{pre}$ steps;

2. *Iterative pruning phase*: We iteratively remove parameters. Each pruning iteration starts by removing a fixed fraction of the remaining parameters, and then training for $P$ gradient steps. This continues for $N_{prune}$ pruning iterations, until the model reaches the desired sparsity $S$.

3. *Sparse recovery phase*: Fixing the sparsity pattern from the previous phase, we further train the sparse model for $N_{post}$ steps to recover any accuracy that is lost due to pruning.

Based on Frantar et al. (2024); Zhu & Gupta (2017); Bambhaniya et al. (2024), we fix $P = 100$.

Given a starting dense parameter count, a target sparsity $S$, and a compute budget, we systematically search for the optimal allocation of compute across these three phases, $N_{pre}$, $N_{prune}$ and $N_{post}$ within a grid of possible allocations, to achieve the best evaluation loss in Section 6.

**Effective compute.** Following (Kaplan et al., 2020; Hoffmann et al., 2024), we approximate the total compute as 6 times the number of parameters multiplied by the number of training tokens. We scale this linearly with sparsity to report effective compute for sparse models (Frantar et al., 2024), noting that our implementation masks pruned parameters rather than eliminating their computation.

**Chinchilla scaling law.** Neural scaling laws model changes in final validation loss under the growth of parameters, data, compute, etc. One widely used scaling law is the Chinchilla scaling law by Hoffmann et al. (2024). This law models the relationship between the loss $L$, the number of parameters $N$, and the number of training tokens $D$ using the following equation:

$$L(N, D) = \frac{A}{N^\alpha} + \frac{B}{D^\beta} + E, \tag{1}$$

where $A$, $B$, $\alpha$, $\beta$, and $E$ are free parameters: $A$ and $\alpha$ describe how loss decreases with increasing model size $N$, while $B$ and $\beta$ describe how loss decreases with increasing training tokens $D$. The constant $E$ represents an irreducible loss. Scaling laws allow practitioners to optimize training configurations to fit within their compute budgets without running costly experiments (Sardana et al., 2024). The laws also hold theoretical value, as they capture the dynamics of how loss changes with data and parameter scaling. We present an overview of existing sparse scaling law in Appendix A.

## 4   METHODS

**Models.** We pre-train both sparse and dense models with starting parameter count ranging from 58M to 468M. We use the LLaMA 2 (Touvron et al., 2023) base model architecture. For each unique model size, we train two versions: one using over 10x the number of tokens corresponding to Chinchilla optimal, and the other using over 20x Chinchilla optimal. We train substantially past Chinchilla-optimal dataset size, following prevailing practice (Touvron et al., 2023; Grattafiori et al., 2024) which benefits inference efficiency.

Table 1 provides model details. Each model name contains the parameter count and a suffix indicating training duration: "10x" means training tokens exceed 10x Chinchilla-optimal, and "20x" means they exceed 20x (Hoffmann et al., 2024). We prune only linear layer parameters, leaving embedding and normalization layers dense. For each dense model, we list its total training tokens and the ratio of tokens to prunable parameters (in parentheses). We train four sparse variants for each dense model, matching their training compute while varying sparsity from 20% to 80%.

| Name | # Prunable | # Tokens |
|------|-----------|----------|
| 58M-10x | 42M | 14.7B (205x) |
| 58M-20x | 42M | 29.4B (409x) |
| 162M-10x | 138M | 33.5B (207x) |
| 162M-20x | 138M | 67.1B (414x) |
| 468M-10x | 435M | 94.4B (217x) |
| 468M-20x | 435M | 188.8B (434x) |

Table 1: Training details for models by size and token count. Numbers in parentheses show the token-to-prunable parameter ratio.

**Dataset.** We use the 'en' partition of the C4 dataset (Raffel et al., 2020). Using the LLaMA 2 tokenizer, this dataset can be tokenized to 197.71 billion tokens.

**Software and hardware.** Our work uses TPUv4 and TPUv5 hardware for training LLMs. We modify MaxText (Davidow et al., 2024) to support sparse pre-training of LLMs.

## 5   SCALING ANALYSIS

While Equation (1) effectively models dense pre-training loss, we show that by changing the parameter count term to average parameter count, the same functional form extends to model sparse pre-training. Here, we derive this modified scaling law analytically and validate it empirically.

### 5.1   A UNIFIED SCALING LAW THAT MODELS DENSE AND SPARSE PRE-TRAINING

Let $T$ denote the total number of pruning iterations, where each iteration consists of a parameter removal step, followed by training at that fixed sparsity. Thus, we may represent a sparse pre-training run as a sequence $(N_1, D_1), (N_2, D_2) \ldots, (N_T, D_T)$, where $N_k$ is the number of remaining parameters, and $D_k$ is the number of tokens at iteration $k \in \{1, \ldots, T\}$. Let $\bar{N}$ denote the average parameter count during training, i.e., $\bar{N} = \frac{1}{T} \sum_{k=1}^{T} N_k$. When $N_k = N_{k'}$ for all $k, k'$, we recover dense training, where the number of parameters does not change throughout training, and $\bar{N} = N_1$.

We model the relationship between the final evaluation loss $L$, the average number of model parameters $\bar{N}$, and the total number of training tokens using the following equation:

$$L(N, D) = \frac{A}{\bar{N}^\alpha} + \frac{B}{D^\beta} + E, \tag{2}$$

where $D$ is the total number of training tokens, and $A$, $B$, $E$, $\alpha$, and $\beta$ are free positive parameters. Importantly, our proposed scaling law retains the same functional form as the Chinchilla scaling law in Equation (1), but replaces the number of dense parameters $N$ with the average parameter count $\bar{N}$. When sparsity is set to 0, our scaling law reduces to the original Chinchilla scaling law.

### 5.2   THEORETICAL ANALYSIS: AVG PARAM COUNT AS EFFECTIVE MODEL SIZE

Using certain assumptions—either standard in prior work or validated through empirical observations, we provide an analytical derivation demonstrating that the average parameter count in sparse pre-training plays the equivalent role of the fixed parameter count in dense training. This motivates unifying both pre-training techniques under the same scaling law as shown in Equation (2).

**Assumptions.** Our analysis rests on two key assumptions: one empirically justified and one extensively used in prior work:

1. Log loss decays linearly with log compute for fixed parameter count pre-training;
2. Total compute for processing a fixed number of tokens is proportional to the number of active parameters in the model.

Kaplan et al. (2020) presented empirical evidence for Assumption (1) when fixed parameter count pre-training falls within the "compute optimal" regime, meaning that the model is appropriately sized to fully utilize the available compute. In our experiments, we extensively tune our sparse pre-training configurations to bring us close to this regime. In this setting, it is known that the loss $L$ evolves as a function of the training compute $C$ as

$$L(C) = (A/C)^{\alpha}, \tag{3}$$

where $L(C)$ represents the loss at compute $C$, and $\alpha > 0$ is a constant that governs the rate of loss decay as compute increases. We apply this relationship to model loss decay within each pruning iteration, where parameter count stays fixed.

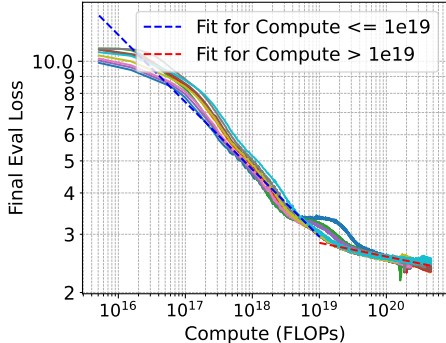 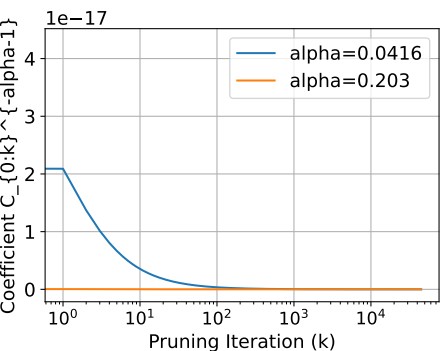

Figure 2: **Left:** Loss vs effective compute for 410M models for various sparsity levels and training durations. **Right:** Estimated $\alpha$ coefficient in the scaling law for 410M model.

Assumption (2) has been heavily used in previous scaling law work (Kaplan et al., 2020; Sardana et al., 2024; Frantar et al., 2024), which model compute as proportional to the number of parameters, batch size, and number of iterations. Since we use a fixed number of tokens per step, the compute per step is directly proportional to the number of active parameters at that step.

**Loss modeling.** To derive the average parameter scaling law, we start with a Taylor series expansion of the loss, as modeled by Equation (3) (Assumption 1), at pruning iteration $k$, where $C_{0:k}$ denotes the cumulative training compute up to iteration $k$. This yields an approximation for the change in loss due to an incremental increase in compute $\Delta C_k$:

$$\Delta L_k \approx -\alpha A^{\alpha} C_{0:k}^{-\alpha-1} \Delta C_k. \tag{4}$$

Applying the above approximation across all pruning iterations, and relying on Assumption 2, we express the total changes in loss over $T$ training steps as proportional to

$$\Delta L_{\text{total}} = \sum_{k=1}^{T} \Delta L_k \propto \sum_{k=1}^{T} C_{0:k-1}^{-\alpha-1} \times N_k, \tag{5}$$

where we have used the fact that computation is known to be linear in the number of parameters (Kaplan et al., 2020) (Assumption 2): $\Delta C_k \propto N_k$.

For realistic values of $C$ and $\alpha$, we find that the terms $C_{0:k-1}^{-\alpha-1}$ remain very stable. In Figure 2 (right), we plot $C_{0:k-1}^{-\alpha-1}$ as a function of pruning iterations for the 410M model experiments, with

empirically fitted $\alpha$ values of 0.041 and 0.203. After about 100 pruning iterations, we observe that this coefficient becomes essentially constant. Further, note that this sum of loss decreases does not take into account the temporary increase in loss due to pruning at a specific step. This is justified as we have observed empirically that, during training, the loss spikes at the pruning step do not significantly affect the final pre-training loss. One may observe this from Figure 2 (left), where we present the loss curves for sparse pre-training 410M models. Instead, the loss only depends on the number of non-zero parameters and on the amount of computation during the iteration.

**Effective model size.** Summing across all $T$ iterations, we find that the total change in loss $\Delta L$ is proportional to the average parameter count during sparse pre-training. Taking the log of this relationship shows that the log change in loss varies linearly with log average parameter count. This demonstrates that sparse pre-training follows the same log-linear relationship between loss and model size as dense scaling Equation (3) suggests, motivating us to extend established dense scaling principles to sparse settings by using average parameter count as effective model size.

**Further validation.** Our analysis predicts that sparse pre-training, despite its varying parameter count, should maintain the same log-linear relationship between loss and training compute as dense pre-training. Figure 2 (left) shows the relationship between the training loss of sparsely pretrained 410M models and effective training compute consumption. The data confirms that the log-linear relationship largely holds. We empirically observe a transition point around $10^{19}$ floating point operations, corresponding to the first 2.6% of training steps. A linear fit on the loss data before and after this point estimates the scaling parameter $\alpha$ to be approximately 0.041 and 0.203, respectively.

### 5.3 EMPIRICAL ANALYSIS

In this subsection, we fit our proposed scaling law with empirical data.

**Fitting method.** We optimize key aspects of our sparse pre-training configuration, including the learning rate, batch size, and compute allocations across the three stages of sparse pre-training (see Section 6 for details). We fit our proposed scaling law (Equation (2)) using the final evaluation loss obtained from sparse pre-training experiments with these optimal configurations. Our experiments cover 5 sparsity levels (0%, 20%, 40%, 60%, 80%), 3 model sizes (58M, 162M, 468M), and 2 training durations (10x Chinchilla optimal and 20x Chinchilla optimal), producing 30 data points.

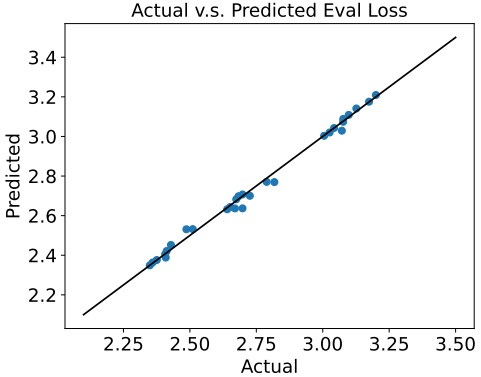

Figure 3: Predicted eval loss from our fitted scaling law versus the actual achieved final loss.

Following the methodology in Hoffmann et al. (2024), we used the L-BFGS algorithm (Liu & Nocedal, 1989) and a Huber loss with $\delta = 1 \times 10^{-3}$ to improve robustness against outliers. We set the maximum number of L-BFGS iterations to 1000, which we empirically find suitable for ensuring convergence. We initialized the scaling law's free parameters with the same random values as in Hoffmann et al. (2024). To account for possible local minima, we sampled 100 initializations from the random grid and selected the parameters with the best Huber loss, following the precedent in Hoffmann et al. (2024); Frantar et al. (2024).

**Results.** We present the predicted model evaluation loss and the actual final evaluation loss in Figure 3. Across different model sizes and training durations, our fitted scaling law models the final model loss with sufficient accuracy. Specifically, the average absolute difference between the predicted and actual loss is 0.016. The distribution of prediction error varies across sparsity levels: the maximum mean absolute difference occurs at 60% sparsity with 0.03, while the minimum occurs at 0% sparsity with 0.007. We attribute this disparity to the scaling analysis not fully accounting for the regularization effects of sparsity (Jin et al., 2022).

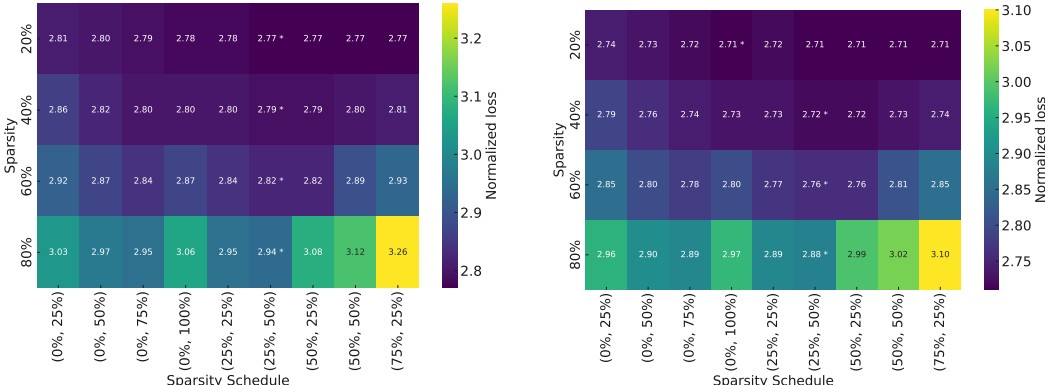

Figure 4: Optimal sparsity schedule sweep for 162M-10× (left) and 162M-20× (right) models. Each tuple on the x-axis, $(t_d, t_s)$, represents the percentage of training time spent for dense traning $(t_d)$, and percentage of time spent gradually pruning $(t_s)$.

**Conclusion.** By replacing the parameter count term with an average parameter count in Chinchilla scaling law (Equation (1)), we demonstrate that this modified scaling law, presented in Equation (2), can effectively predict the final evaluation loss of sparse pre-training.

# 6 SEARCHING FOR THE OPTIMAL SPARSE PRE-TRAINING CONFIGURATION

In Section 3, we introduced the three phases of sparse pre-training—dense training, iterative pruning, and sparse recovery. We now examine how to optimally allocate compute across these pre-training phases. In this section, we present our experimental search for optimal sparse pre-training configurations; we used these searched optimal configurations to validate our scaling law in Section 5.

## 6.1 SPARSE PRE-TRAINING CONFIGURATION SWEEP.

**Sparsity schedule sweep.** For each combination of initial dense parameter count, target sparsity, and training compute budget, we optimize compute allocation across the three training phases (Section 3) to maximize final model evaluation loss.

A large body of research shows that pruning too early can trap models in suboptimal minima (Gale et al., 2019; Frankle et al., 2019; Paul et al., 2023; Bambhaniya et al., 2024; Lu et al., 2023). These findings suggest that one needs to allocate substantial compute to the dense training phase before iterative pruning phase may begin. Similarly, research also shows that removing too many weights in one iteration leads to poor final model quality (Renda et al., 2020). Therefore, reaching target sparsity requires a sufficiently large number of pruning iterations, each removing only a moderate fraction of weights. These competing demands make sparsity schedule design challenging.

Focusing on the 162M-10× and 162M-20× models, we systematically search for the optimal sparsity schedule by evaluating all valid combinations of compute allocated to the dense training phase (0%, 25%, 50%, 75% of total training FLOPs) and the weight removal phase (25%, 50%, 75%, 100%). For a schedule to be valid, the combined compute for the dense training and weight removal phases must not exceed 100%, with the remaining compute allocated to the sparse recovery phase. For this sweep, we adopt the same hyperparameter configurations (learning rate, batch size, etc.) used for equivalent model configurations in the Pythia suite of models (Biderman et al., 2023).

**Sparsity Schedule Results.** We present the results of our sparsity schedule sweep in Figure 4. We encode each schedule on the x-axis with a 2-tuple. The first value represents the proportion of compute allocated to the dense training phase, and the second value represents the compute allocated to the weight removal phase. Our results consistently show that allocating 25% of total compute to the dense training phase and 50% to the weight removal phase yields either the optimal training loss or a result within 0.01 of the minimum.

**Learning rate and batch size sweep.** We subsequently investigate the optimal learning rate and batch size to use for sparse pre-training. We sweep through a grid of $[0.0004, 0.0016, 0.0064]$ for learning rate and $[0.125M, 0.5M, 2M]$ for batch size.

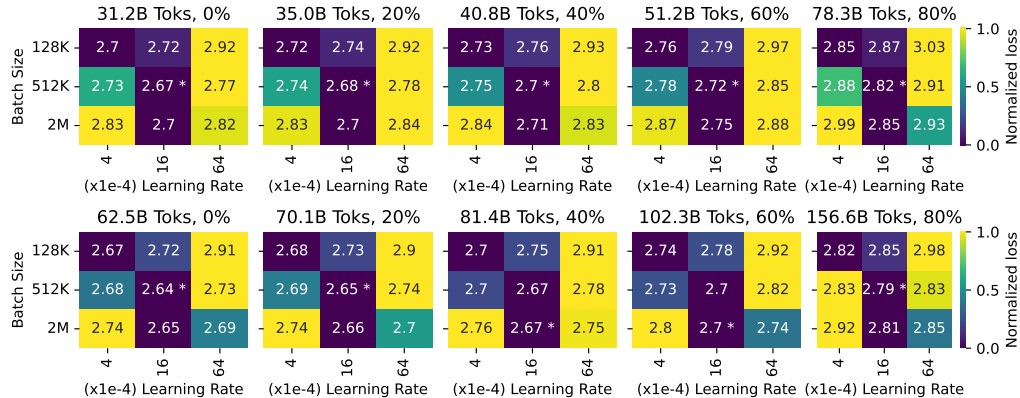

Figure 5: Batch size and learning rate sweep for 162M models.

**Learning rate and batch size results.** We present our learning rate and batch size sweeps in Figure 5. Our findings consistently show that using the optimal hyper-parameters for dense pre-training either achieves the optimal evaluation loss for sparse pre-training or comes very close (within 0.01 difference).

## 6.2 CLOSER LOOK AT PRUNING SCHEDULE

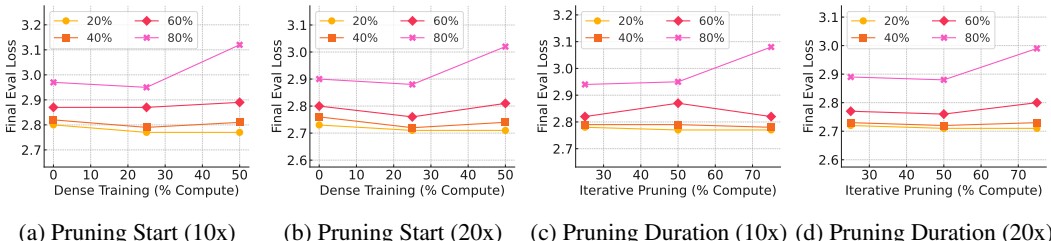

(a) Pruning Start (10x)   (b) Pruning Start (20x)   (c) Pruning Duration (10x) (d) Pruning Duration (20x)

Figure 6: A closer look at failure modes for non-optimal sparsity schedules.

In this section, we visualize the impact of allocating different fractions of the total compute to the dense training and iterative pruning phases on final evaluation loss.

**Dense training compute.** To determine the optimal fraction of compute to allocate to the dense pre-training phase, we vary the dense compute between 0%, 25%, and 50%. Throughout these experiments, we keep the iterative pruning phase compute fixed at 50%, as this was previously found to be optimal, and adjust the sparse recovery phase compute to maintain a constant total compute for sparse pre-training. The results are shown in Figures 6a and 6b.

Our findings indicate a clear optimal allocation of dense training compute at 25% of total compute, where 7 out of 8 cases reach their lowest loss. This trend is consistent across different sparsity levels (ranging from 20% to 80%) and both training regimes (10x and 20x Chinchilla-optimal).

Additionally, the results suggest that allocating too much compute to dense training (50%) leads to worse final loss, particularly for high-sparsity models. This underscores the importance of allocating sufficient compute to the sparse recovery phase.

**Iterative pruning compute.** We explore the effect of varying the compute allocation to the iterative pruning phase. First, we fix the dense training compute at 25%, as this was previously found to be optimal. Then, we vary the pruning duration while adjusting the sparse recovery phase to keep the total compute constant. We visualize these results in Figures 6c and 6d.

We find that the optimal allocation of iterative pruning compute varies across training regimes. For the 10x Chinchilla model, the lowest evaluation loss occurs with a 25% allocation to iterative prun-

ing (2.83), while for the 20x Chinchilla model, the optimal allocation is 50% (2.77). Despite these differences, allocating 50% of total compute to iterative pruning consistently results in reasonable evaluation loss across both regimes. Interestingly, extending the iterative pruning allocation beyond 50% tends to degrade performance, particularly in high-sparsity models (80%). These findings suggest that high-sparsity models benefit most from moderate allocations to iterative pruning, ensuring sufficient compute for sparse recovery after weight removal.

## 7 CONCLUDING REMARK

Our work examines sparse pre-training for large language models (LLMs) and presents a unified scaling law that effectively models both sparse and dense scaling.

**Value of sparse pre-training.** Sparsely pre-trained LLMs match the final evaluation loss of dense models when their average parameter counts are the same (Figure 1). Unlike dense pre-training, which maintains a constant parameter count, sparse pre-training starts with more parameters and gradually reduces them, effectively redistributing parameter count over training steps. For the same average parameter count, this redistribution does not change total training compute. Within the model sizes, sparsity levels, and schedules we explored, redistributing parameters maintains the same training compute and evaluation loss trade-off as dense pre-training while achieving smaller final model size. This makes sparse pre-training valuable: it improves the three-way trade-off among training compute, evaluation loss, and final model size.

**Scaling prescription.** Chinchilla (Hoffmann et al., 2024) defines compute-optimal LLM training configurations by minimizing training compute alone, yet practical LLMs like LLaMA (Touvron et al., 2023; Grattafiori et al., 2024) often train longer to reduce inference costs. Sardana et al. (2024) formalizes this practice by minimizing lifetime compute—the sum of training and inference compute. Applying the same analysis with our scaling law (see Appendix B), we find that training with less data can achieve the same target loss while reducing lifetime compute, compared with the prescription by Sardana et al. (2024). This surprising advantage stems from sparse pre-training's key feature: it decouples the average parameter count during training, which governs model quality, from the final parameter count after training, which determines inference compute. This decoupling enables a better balance between training efficiency and inference costs, making sparsity a crucial factor in designing compute-optimal training configurations for LLMs.

**Compression rate.** Our findings suggest that up to a certain compression rate, sparse pre-training compresses the LLM during training without loss in quality. For a given sparse pre-training configuration, we define its compression rate as the ratio between two model sizes: the smallest dense model that matches the sparse model's evaluation loss (while keeping other factors like compute, data constant) and the final sparse model size. From our analysis, we know that a dense model needs as many parameters as the sparse model's average parameter count during pre-training to match its quality. Therefore, for a sparse pre-training configuration, its compression rate is also the ratio between average and final parameter counts. Within the LLMs we explore, we reach the maximum compression rate at 80% final sparsity, where our sparsity schedule results in an average parameter count of about 40% of the initial dense parameter count, yielding a 2x lossless compression rate.

**Limitation.** We note that, due to the lack of adequate software and hardware support for executing matrix multiplications with unstructured sparsity, we are unable to demonstrate computational savings from sparse pre-training. For the same reason, the scale of our study is limited, as the actual compute required may be up to $1/(1 - \text{sparsity})$ times the effective compute we estimate for sparse pre-training. For experiments with 80% sparsity, this factor reaches 5x.

**Conclusion.** In this work, we unify sparse and dense pre-training through a single scaling law that uses average parameter count, showing its effectiveness in modeling evaluation loss across model sizes, sparsity levels, and training durations. Our analysis reveals several practical benefits: sparse pre-training reduces inference costs without sacrificing model quality or increasing training compute; compared to existing dense pre-training prescriptions, incorporating sparsity maintains model quality while reducing both training data and lifetime compute costs. Together, our work establishes sparse pre-training as a promising alternative to dense pre-training.

**Reproducibility statement.** We have detailed our dense LLM training configurations in Section 4, along with our hardware and software setup. Additionally, in Section 6, we provide an extensive discussion of the pruning configurations we explored and identified.

**Acknowledgment.** We are deeply grateful to Elias Frantar, Naveen Kumar, Sanjiv Kumar, Daniel M. Roy, and Clemens Schaefer for their valuable feedback and thoughtful review of this paper. We also acknowledge the critical support provided by the Google CoreML Performance Team, and Google Research during this project. We further recognize the extended team at Google DeepMind, who enabled and supported this research direction.

This work was in part supported by the Sloan Foundation, the MIT-IBM Watson AI Lab, Apple, and SRC JUMP 2.0 (CoCoSys).

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

## A    EXISTING SPARSE SCALING LAW

Frantar et al. (2024) propose a model for the evaluation loss of a sparsely pre-trained LLM based on the final sparsity $S$, the number of non-zero parameters $N$ at the end of pre-training, and the amount of training data $D$, as follows:

$$L(S, N, D) = \left(a_S(1 - S)^{b_S} + c_S\right) \cdot \left(\frac{1}{N}\right)^{b_N} + \left(\frac{a_D}{D}\right)^{b_D} + c$$

Compared to the Chinchilla scaling law, this model introduces the following modifications:

1. A sparsity term $a_S(1 - S)^{b_S}$, which introduces two new sparsity-specific parameters, $a_S$ and $b_S$, allowing the model to account for the effect of sparsity on loss.

2. Instead of using the total number of parameters $N$ as in dense training, the model uses the number of non-zero parameters remaining at the end of sparse pre-training.

## B    SPARSE SCALING SUGGESTS TRAINING WITH LESS DATA

Following the methodology of Sardana et al. (2024), we compute the recommended dataset size and model size to achieve a specific model loss according to Chinchilla Hoffmann et al. (2024),

Beyond Chinchilla Sardana et al. (2024), and our proposed scaling laws. We find our proposed sparse pretraining technique recommends using less data.

For a target training loss of 1.89, and a total expected inference traffic of 100T tokens, we have these recommendations:

- **Chinchilla:** Optimizing for training compute, Chinchilla scaling law recommends training a 70B model for 4.26T tokens, the model consumes $15.8 \times 10^{24}$ FLOPs over its lifetime.

- **Beyond Chinchilla:** Optimizing for training and inference compute, Chinchilla scaling law recommends training a 23.5B model for 24.4T tokens. With 100T inference tokens, the model consumes $8.14 \times 10^{24}$ FLOPs over its lifetime.

- **Ours:** Optimizing for training and inference compute and fixing sparsity level at 80%, our proposed scaling law recommends training a model with average parameter count of 28.0B, resulting in a 14.0B model at the end of pretraining for 16.6T tokens. With the same 100T inference tokens, the model consumes $5.58 \times 10^{24}$ FLOPs over its lifetime, resulting in a 31.4% additional lifetime compute saving. This number grows even larger with increased inference traffic, since the benefit of sparse pretraining lies predominantly in inference compute saving.

Compared with Sardana et al. (2024), our prescription achieves the same target loss while using less data and less compute, suggesting that sparse pretraining is more data efficient.

The intuition behind our counter-intuitive prescription is that sparse pretraining decouples average parameter count during training (which correlates with quality) from inference parameter count (which correlates with inference compute). This decoupling allows us to train models with larger average parameter counts while still producing a smaller model for inference compared with traditional dense scaling.

## C VALIDATION USING LARGER MODELS

In Figure 1, we demonstrated that sparse and dense pre-training runs, starting with 138 million prunable parameter models and matching average parameter counts, achieve similar perplexity. This section extends the same analysis to models exceeding 1 billion parameters to evaluate the generality and robustness of these observations.

**Method.** We pre-trained a sparse model starting with 1.14 billion parameters, pruning it to 50% sparsity over 144 billion tokens. The corresponding dense model was configured to match the sparse model's average parameter count during training. Both models used the same compute budget of $6.8 \times 10^{20}$ FLOPs, approximately 5x the Chinchilla-optimal budget for a 1B-parameter model. The sparse model uses global iterative magnitude pruning, while the dense model followed a standard training setup.

**Results.** The sparse model achieved a final perplexity of 2.44, closely matching the dense model's 2.46. At the end of training, the dense model has 1.34 times as many parameters as the sparse model, highlighting that sparse pre-training achieves significant parameter reduction while maintaining comparable performance.

**Conclusion.** These results confirm that sparsely pre-trained language models with matching average parameter counts achieve comparable perplexity to their dense counterparts, even at a larger scale. This consistency demonstrates the generality and robustness of our observations.

## D DOWNSTREAM TASK PERFORMANCE

While perplexity is an important metric, downstream task performance is critical for understanding the practical utility of sparse pretraining. This section evaluates the pair of sparse and dense models produced in Appendix C on various downstream benchmarks to assess their task performance.

**Method.** To evaluate downstream task performance, we selected four commonly used benchmarks that test reasoning and commonsense knowledge:

- **PIQA (Physical Interaction Question Answering):** This dataset evaluates a model's ability to reason about everyday physical commonsense, such as choosing the most plausible way to perform a physical task.

- **ARC (AI2 Reasoning Challenge):** ARC consists of grade-school science questions designed to test reasoning and application of scientific knowledge, divided into *Easy* and *Challenge* subsets.

- **Lambada:** This dataset tests a model's ability for long-range coherence by requiring it to predict the last word of a passage, where understanding the full context is critical.

- **Winogrande:** This dataset evaluates commonsense reasoning by asking the model to resolve pronoun references in ambiguous sentences.

We evaluated both the sparse model (1.14B parameters, 50% sparsity) and the dense model (matching average parameter count) on the aforementioned datasets.

**Results.** As shown in Table 2, the sparse and dense models achieved nearly identical performance across all benchmarks. The sparse model's average geometric mean accuracy was 42.42%, compared to 42.45% for the dense model, with no significant difference observed.

Table 2: Downstream task evaluation results for sparse and dense models.

| Model | PIQA | ARC Easy | ARC Challenge | Lambada | Winogrande | Geomean |
|---|---|---|---|---|---|---|
| 50% Sparse | 69.7 | 42.05 | 24.74 | 36.52 | 51.85 | 42.42 |
| Matching Dense | 69.26 | 41.58 | 26.37 | 34.62 | 52.41 | 42.45 |

**Conclusion:** These results demonstrate that sparsely pre-trained language models are equally capable as their dense counterparts in downstream tasks when matched on average parameter count, highlighting the practical utility of sparse pretraining.

