# OpenReview forum: "The Journey Matters: Average Parameter Count over Pre-training Unifies Sparse and Dense Scaling Laws"
_ICLR.cc/2025/Conference — ICLR 2025 Poster_

### Official Review · Reviewer_Kf8Y · 2024-11-04

**Soundness:** 3
**Presentation:** 4
**Contribution:** 3
**Rating:** 6
**Confidence:** 4

**Summary:**

The paper studies the scaling law for pretraining sparse language models. It argues that sparse pretraining reduces both pretraining and inference costs since the final model can be smaller and requires less computation during inference. The authors modify the Chinchilla scaling law for dense models to use the average number of active parameters (receiving gradients) in training. The modified scaling law unifies both sparse and dense models.  They also conduct extensive studies to search for optimal sparse pretraining configurations for LLMs. The key findings include pruning starting at 25% and ending at 75% obtains near optimal final eval loss, optimal learning rates, and batch sizes for sparse training match dense models under the same compute budget. The paper further provides both theoretical justifications and empirical validation for the scaling law and presents a simple recipe for sparsely pretrained models.

**Strengths:**

- The paper studies an interesting research problem, the scaling law for pretraining sparse models. It develops a modified scaling law that unifies both dense and sparse models.
- The analysis and empirical validation are solid and convincing. The findings are supported by both theoretical justifications and experiments over 80 sparse pretraining configurations. The optimal settings for pretraining sparse models provide valuable practical lessons.
- The paper itself is well-written, clearly structured, and easy to follow.

**Weaknesses:**

- One major concern is that the scaling law for dense models seems to have less impact, especially given that generally more data and compute resources are better to obtain better and more capable LLMs, LLaMA 2 and LLaMA 3 papers provide such evidence. How would unifying the scaling law for both dense and sparse models be useful under such a context?
- The paper mainly focuses on LLaMA style model architecture for sparse pretraining, any discussion on other sparse architectures like the mixture of experts (MoE)? Otherwise, it may be a bit overclamied.
- The other concern is that previous scaling laws at least study model parameters with 1B scale, but this paper only discusses models under 500M parameters, making the claim of scaling law for LLM weaker.
- While the paper acknowledges the limitation of evaluation loss, it would be very useful to report the performance numbers of downstream tasks, especially given the size of models is small. No compute limitation prohibits such evaluation.

**Questions:**

See weaknesses.

---

> ### Author Response · Authors · 2024-11-25
> **Author Response**
>
> We thank the reviewer for the constructive feedback and thoughtful questions.
>
> Below we share our responses:
>
> > One major concern is that the scaling law for dense models seems to have less impact, especially given that generally more data and compute resources are better to obtain better and more capable LLMs, LLaMA 2 and LLaMA 3 papers provide such evidence. How would unifying the scaling law for both dense and sparse models be useful under such a context?
>
> Developing unified scaling laws for dense and sparse models have both theoretical and practical interests. Theoretically, it suggests that architectural changes due to sparsity do not change the rate of convergence of excess risk (sample complexity). In other words, for the data we model in practical scenarios, theorists should not expect to show that learning rate improves in terms of the samples needed with sparsity. Practically, it provides a valuable tool for practitioners to understand how to effectively transition their existing dense pre-training configurations to sparse ones. It also helps in determining the optimal amount of training computation, accounting for downstream inference cost as in [1].
>
> [1] Beyond Chinchilla-Optimal: Accounting for Inference in Language Model Scaling Laws, ICML 2024.
>
> > The paper mainly focuses on LLaMA style model architecture for sparse pretraining, any discussion on other sparse architectures like the mixture of experts (MoE)? Otherwise, it may be a bit overclamied.
>
> Thank you for bringing up this interesting question on Mixture of Experts models! We highlight that MoE-type sparsity are coarse-grained whereas the parameter sparsity we study are fine-grained; thus the introduction of these different types of sparsity may affect training dynamics differently.
>
> For this reason, we believe other forms of sparsity such as MoE are outside of the scope of our study. We are happy to revise our paper to emphasize that the focus of our study is parameter sparsity.
>
> We have carefully reviewed and revised our paper as follow:
>
> - In the abstract we begin by writing “**Parameter pruning** has emerged as a promising technique to address…”
> - In the introduction when we introduce sparse training, we rephrased to explain that sparse training “reduces the computational burden by using only **a subset of the neural network parameters** both during training and inference.…”.
> - In the conclusion, we revised our opening statement to again emphasize our focus on parameter sparsity: “Our work examines pre-training large language models with **parameter sparsity** and presents a unified scaling law that effectively models both sparse and dense scaling.”
>
> Again, we are happy to make additional clarifications.

---

> > ### Comment · Reviewer_Kf8Y · 2024-11-25
> >
> > Thank you for the rebuttal. I still have concerns about the value of scaling laws in practical settings, not specific points like learning rates or architectural changes. My point is that more data and compute resources lead to better and more capable LLMs. Are there any evidence or explanations in the proposed scaling law for sparse models that refute the argument?
> >
> > Regarding MoE, I understand that a detailed comparison is probably out of scope, thank you for clarifying this in writing. However, I was expecting some discussion on how the current scaling law for sparse models would apply to MoE models.

---

> > > ### Author Response · Authors · 2024-11-27
> > > **Practical value of scaling laws**
> > >
> > > We thank the reviewer for their responsiveness and for raising this insightful question about scaling laws.
> > >
> > > ---
> > >
> > > ## On "More Data and Compute Resources Lead to Better and More Capable LLMs"
> > >
> > > The combination of more data and more compute does not universally guarantee better model quality. For a counter-example, using the Chinchilla scaling law [1] and parameters from [2], we observe the following:
> > >
> > > - **Configuration 1**: A 1-billion-parameter model, trained on 20 billion tokens, using $1.2 \times 10^{20}$ FLOPs, achieves a final loss of **2.53**.
> > > - **Configuration 2**: A 100-million-parameter model, trained on 400 billion tokens, using $2.4 \times 10^{20}$ FLOPs, achieves a final loss of **2.73**.
> > >
> > > Despite consuming twice the compute and 20x the data, Configuration 2 produces a less capable model than Configuration 1. This example demonstrates the original motivation behind scaling laws: guiding the optimal allocation of **training compute** to model size and training data, thereby avoiding inefficient configurations such as Configuration 2. We discuss extending scaling laws to **inference compute** shortly below.
> > >
> > > ---
> > >
> > > ## On the Relevance of Scaling Laws in Contemporary Practical LLMs
> > >
> > > The reviewer raises an important point about current practices, where models such as LLaMA 1/2/3 are often trained on far more data than recommended by Chinchilla. However, we believe scaling laws remain highly relevant to guide their training configurations, particularly when both training and inference costs are considered. As shown in [3], scaling laws can extend to jointly optimize training and inference compute over a model's lifetime.
> > >
> > > ### Optimizing Total Compute:
> > > For a target loss $L(N, D_{train})=l$, we aim to find optimal $N^*, D_{train}^*$ that minimize total compute $C$, which is the sum of training and inference compute: $C = 6 \cdot N \cdot D_{\text{train}} + 2 \cdot N \cdot D_{\text{inference}}$
> > > - $ 6 \cdot N \cdot D_{\text{train}} $: Training compute (forward and backward passes).
> > > - $ 2 \cdot N \cdot D_{\text{inference}} $: Inference compute (forward passes only), where $D_{\text{inference}}$ is the total number of tokens users may sample from the LLM over its lifetime. This quantity may be estimated from historical data as in [3].
> > >
> > > This optimization problem accounts for both training and inference costs while achieving the desired model quality. Solving it often recommends training smaller models for longer compared to Chinchilla's original recommendations, consistent with standard practices of LLaMA training.
> > >
> > > ---
> > >
> > > ## How Our Results Influence This Optimization Problem
> > >
> > > Our work extends this framework to model sparse pre-training. Specifically:
> > >
> > > 1. **Transition from dense to sparse pre-training**: For the same optimal dense model size $ N^* $ and training tokens $ D^*_{\text{train}} $ that optimizes the aforementioned equations, a sparse pretraining schedule can reduce the *effective inference size* $ N_{\text{inference}} $ -- the number of active parameters used during inference after sparse pre-training. One may configure the average parameter count $ N_{\text{average}} $ during training to be $ N^* $, ensuring constant training compute (ignoring overhead of sparse matrix operations for simplicity). However, the smaller $ N_{\text{inference}} $ at the end of sparse-pretraining can substantially reduce inference compute:
> > >    $
> > >    C_{\text{inference}} = N_{\text{inference}} \cdot D_{\text{inference}} < N^* \cdot D_{\text{inference}}
> > >    $
> > >
> > > 2. **Re-framing Optimization Problem**: One can also select a target sparsity level based on hardware support, say 50%, and minimize the total compute under the same quality target $l$ as:
> > >    $
> > >    C = 6 \cdot N_{\text{average}} \cdot D_{\text{train}} + 2 \cdot N_{\text{inference}} \cdot D_{\text{inference}}.
> > >    $
> > >    Using our proposed sparse-pretraining configuration, at 50\% sparsity, $N_{\text{inference}} \approx 0.75 \cdot N_{\text{average}} $, yielding:
> > >    $
> > >    C = 6 \cdot N_{\text{average}} \cdot D_{\text{train}} + 2 \cdot 0.75 \cdot N_{\text{average}} \cdot D_{\text{inference}}.
> > >    $
> > >    We can thus solve for $ N_{\text{average}} $ and $ D_{\text{train}}$ that minimizes the sum of training and inference compute constraints for a particular quality target.
> > >
> > > By offering a principled formula to reason about sparse scaling, sparse pre-training adds importance and practical applicability to scaling laws, particularly in compute-constrained deployment scenarios.
> > >
> > > [1] Training Compute-Optimal Large Language Models, NeurIPS 2022.
> > >
> > > [2] Chinchilla Scaling: A replication attempt, arXiv 2024.
> > >
> > > [3] Accounting for Inference in Language Model Scaling Laws, ICML 2024.
> > >
> > > We will discuss MoE in a separate follow-up post.

---

> ### Author Response · Authors · 2024-11-25
> **Author Response (Continued)**
>
> > The other concern is that previous scaling laws at least study model parameters with 1B scale, but this paper only discusses models under 500M parameters, making the claim of scaling law for LLM weaker.
>
> While we cannot train larger models in the over-training regime explored in the paper (10x Chinchilla optimal and beyond) fast enough to finish within the rebuttal period, we have trained a pair of sparse/dense models with a smaller compute budget and matching average parameter count to demonstrate our finding in Fig.1 scales. As a reminder, Fig.1 shows that with all other factors held constant, the average number of active parameters is an effective predictor of the final evaluation loss for sparse models, just as it is for dense models.
>
> The sparse model starts from a 1.14B model and is pruned to 50% sparsity over 144B tokens and the dense model matches its average parameter count and trains for the same number of tokens. This compute budget (6.8E20 FLOPs) corresponds to ~5x Chinchilla optimal for the 1B model.
>
> With this pair of models, we confirmed that our Fig. 1 result scales: the average parameter count again is strongly predictive of model final evaluation loss. Specifically, the sparse model achieved a perplexity of 2.44, while the matching dense model achieved 2.46. Notably, at the end of pre-training, the matching dense model was 1.34x the size of the sparse model.
>
> > The only evaluation metric being used in this paper is the pretrained model's perplexity (or pretraining evaluation loss) without downstream task evaluations. As models could forget knowledge in pretraining because of pruning, a finer-grained analysis with task-specific revaluations could be beneficial.
>
> We re-evaluate our claims in Fig.1 using downstream task evaluations. We pick tasks that require varying combinations of knowledge recall and reasoning capabilities. We again show that, with other factors held constant, pairs of sparse and dense models trained with matching average parameter count yields similar downstream eval performance. Similar to Chinchilla[4], we focus our evaluation on the largest models trained, which is the 1.14B model with 50% sparsity and its smaller dense model counterpart from our previous answer.
> We used four downstream eval benchmarks:
> - **PIQA (Physical Interaction Question Answering)**: This dataset evaluates a model's ability to reason about everyday physical commonsense, such as choosing the most plausible way to perform a physical task.
> - **ARC (AI2 Reasoning Challenge)**: ARC consists of grade-school science questions designed to test reasoning and application of scientific knowledge, divided into "Easy" and "Challenge" subsets.
> - **Lambada**: This dataset tests a model's ability for long-range coherence by requiring it to predict the last word of a passage, where understanding the full context is critical.
> - **Winogrande**: This dataset evaluates commonsense reasoning by asking the model to resolve pronoun references in ambiguous sentences.
>
> | Model Name           | PIQA  | ARC Easy | ARC Challenge | Lambada | Winogrande | Geomean|
> |----------------------|-------|----------|---------------|---------|------------|------------------|
> | 1.1B 50% Sparse      | 69.7  | 42.05    | 24.74         | 36.52   | 51.85      | 42.42            |
> | Matching Small Dense | 69.26 | 41.58    | 26.37         | 34.62   | 52.41      | 42.45            |
>
> We do not observe systemic differences in the sparse versus dense evaluation results; despite the matching small dense model being 1.34x the size of the sparse model, they both achieve similar average accuracy (42.42% vs 42.45%); in other words, with all other factors held constant, the average number of active parameters is an effective predictor of the downstream task evaluation for sparse models, just as it is for dense models.

---

> > ### Comment · Reviewer_Kf8Y · 2024-11-25
> >
> > Thank you for adding further clarification and downstream task evaluation.

---

> ### Author Response · Authors · 2024-12-02
> **Discussion on MoE Models**
>
> We believe there are several cases to consider when discussing how our proposed sparse scaling laws may apply to MoE models. We preface this discussion by noting that 1. these are hypotheses and educated guesses and 2. as we previously stated, we believe MoE's are quite different from sparsely pre-trained LLMs, they probably require different scaling analysis.
>
> ## Fixed Number of Activated Experts
>
> The case of a fixed number of activated experts is analogous to sparse pre-training with a fixed sparsity with however one key difference - for MoE Fixing the number of activated experts determines the total number of active parameters but does not specify which subset of parameters will be activated whereas for the sparse pre-training algorithm we explore, once weights are pruned, they do not reactivate.
>
> The closest pruning analogy would be advanced pruning algorithms such as RigL [1] and ACDC [2], where pruned weights may rejuvenate. These methods have shown improvements over simpler pruning techniques; therefore, though we speculate that basic principles of sparse-scaling extend similarly to MoE, MoEs probably enjoy better scaling coefficients.
>
> We also hypothesize that using more sophisticated pruning algorithms could yield stronger scaling coefficients for sparse pre-training, too, albeit with added engineering complexity.
>
> We also highlight that there is already study on MoE scaling in such settings [3]. For example, [3] found that the MoE quality improves with increasing number of total experts (in other words, sparsity level) and its benefit over dense models decreases as the model increases its scale. Our results agree with [3] in the benefits of high sparsity level, but we do not observe diminishing returns with the particular model size regime (up to 1B) and pruning algorithm (fine-grained iterative global magnitude pruning) we explore.
>
> ## Dynamic Number of Activated Experts
>
> Some approaches, such as [3], propose dynamically determining the number of activated experts based on task difficulty. In this scenario, conventional scaling analysis becomes challenging because the effective number of parameters varies across training and inference instances.
>
> ## Reducing the Number of Activated Experts
>
> An interesting scenario relevant to our work involves gradually reducing the number of activated experts during the course of pre-training. Although a key difference still exists when compared to sparse pre-training (as MoE does not fix the subset of weights to activate), it raises an intriguing question: could the average number of activated experts during pre-training serve as a predictive metric for MoE performance? Exploring this hypothesis could provide valuable insights into the scaling behaviors of MoE models.
>
> ## Combining Sparsity Techniques
>
> Finally, we emphasize that fine-grained sparsity (as explored in this work) and coarse-grained sparsity (e.g., MoE) can be combined to jointly optimize model compute efficiency. This hybrid approach has potential for further improving scaling behavior and compute efficiency in practical deployments.
>
> [1] Rigging the lottery, ICML 2020.
>
> [2] AC/DC: Alternating Compressed/DeCompressed Training of Deep Neural Networks, NeurIPS 2021.
>
> [3] Unified Scaling Laws for Routed Language Models, ICML 2022.
>
> [4] Harder Tasks Need More Experts: Dynamic Routing in MoE Models, ACL 2024.

---

> > ### Author Response · Authors · 2024-12-02
> > **Follow-up**
> >
> > Dear reviewer Kf8Y:
> >
> > We are eager to know whether our discussion on practical utility of scaling laws and applicability to MoE settings addressed your questions and concerns!
> >
> > Best wishes,
> >
> > Authors of 9129.

---

> > > ### Comment · Reviewer_Kf8Y · 2024-12-02
> > >
> > > Thanks for the clarification. I'm still not convinced by the explanation of sparse scaling laws in practical LLM training, especially given the smaller size of the two configurations. The MoE discussion has addressed my concerns. I would keep the current rating.

---

> > > > ### Author Response · Authors · 2024-12-04
> > > > **Practical Implications of Scaling Laws**
> > > >
> > > > We appreciate your consistently active engagement in the discussion and thoughtful comments!
> > > >
> > > > ## Application of scaling laws in Llama3
> > > >
> > > > We highlight the following use of scaling laws by the Llama 3 team: the flagship Llama-3 405B model was designed with the optimal parameter count and dataset size to achieve compute optimality according to scaling laws. Below is an excerpt from the Llama 3 technical report [1]:
> > > >
> > > > > Llama 3 405B uses an architecture with 126 layers, a token representation dimension of 16,384, and 128 attention heads; see Table 3 for details. This leads to a model size that is approximately compute-optimal according to scaling laws on our data for our training budget of 3.8 × 10^25 FLOPs.
> > > >
> > > > ## Additional counter example to “More data and more compute yields better model”
> > > >
> > > > Furthermore, we analyzed the raw data from Chinchilla scaling law paper from [2], and found many pairs of Efficient/Inefficient training configurations, where the inefficient configuration uses more data and more compute than efficient configurations yet achieves worse loss than the efficient configurations. We highlight 3 of them here. In all cases the inefficient configurations use more compute, more data yet achieved worse (higher) loss. These results highlight the important role of scaling laws to guide optimal allocation of training compute to model parameters versus training data.
> > > >
> > > > | Type        | Training FLOP   | Dataset Size (B)   | Model Size (B)   |   Loss |
> > > > |:------------|:----------------|:-------------------|:-----------------|-------:|
> > > > | Efficient   | 9.729e+20   | 71.03 B          | 2.283 B        |  2.293 |
> > > > | Inefficient | 9.740e+20   | 128.30 B         | 1.266 B        |  2.331 |
> > > >
> > > >
> > > > | Type        | Training FLOP   | Dataset Size (B)   | Model Size (B)   |   Loss |
> > > > |:------------|:----------------|:-------------------|:-----------------|-------:|
> > > > | Efficient   | 9.769e+20   | 61.70 B          | 2.639 B        |  2.286 |
> > > > | Inefficient | 9.775e+20   | 101.20 B         | 1.609 B        |  2.314 |
> > > >
> > > > | Type        | Training FLOP   | Dataset Size (B)   | Model Size (B)   |   Loss |
> > > > |:------------|:----------------|:-------------------|:-----------------|-------:|
> > > > | Efficient   | 1.293e+21   | 72.35 B          | 2.980 B        |  2.276 |
> > > > | Inefficient | 1.297e+21   | 124.90 B         | 1.731 B        |  2.286 |
> > > >
> > > > ## Sparse scaling suggests training with less data
> > > >
> > > > Finally, following the methodology of [3], we compute the recommended dataset size, model size to achieve a specific model loss  according to Chinchilla [4], Beyond Chinchilla [3] and our proposed scaling laws. We find our proposed sparse pretraining technique indeed recommends using less data.
> > > >
> > > > For a target training loss of 1.89, and a total expected inference traffic of 100T tokens, we have these recommendations:
> > > >
> > > > - **Chinchilla**: Optimizing for training compute, Chinchilla scaling law recommends training a **70B model** for **4.26T tokens**, the model consumes **15.8E24 FLOPs** over its lifetime.
> > > > - **Beyond Chinchilla**: Optimizing for training + inference compute, Chinchilla scaling law recommends training a **23.5B model** for **24.4T tokens**. With 100T inference tokens, the model consumes **8.14E24 FLOPs** over its lifetime.
> > > > - **Ours**: Optimizing for training + inference compute, and fixing sparsity level at 80%, our proposed scaling law recommends training a model with average parameter count of **28.0B**, resulting in a 14.0B model at the end of pre-training for **16.6T** tokens. With the same 100T inference tokens, the model consumes **5.58E24 FLOPs** over its lifetime, resulting in a 31.4% additional lifetime compute saving. This number grows even larger with increased inference traffic, since the benefit of sparse pre-training lies predominately in inference compute saving.
> > > >
> > > > Compared with [3], our prescription achieves the same target loss while using less data and less compute, suggesting that sparse pre-training is more data efficient.
> > > >
> > > > The intuition behind our counter-intuitive prescription is that sparse pre-training decouples average parameter count during training (which correlates with quality) from inference parameter count (which correlates with inference compute). This decoupling allows us to train models with larger average parameter count while still producing a smaller model for inference, compared with traditional dense scaling.
> > > >
> > > > [1] The Llama 3 Herd of Models. arXiv 2024.
> > > >
> > > > [2] Chinchilla Scaling: A replication attempt, arXiv 2024.
> > > >
> > > > [3] Beyond Chinchilla-Optimal: Accounting for Inference in Language Model Scaling Laws, ICML 2024.
> > > >
> > > > [4] Training Compute-Optimal Large Language Models, NeurIPS 2024.

---

### Official Review · Reviewer_tYDa · 2024-11-04

**Soundness:** 3
**Presentation:** 3
**Contribution:** 2
**Rating:** 6
**Confidence:** 3

**Summary:**

This paper investigates sparse pre-training as an alternative to post-training pruning for large language models (LLMs) and proposes a modified scaling law based on the average active parameter count. Through extensive experimentation with over 80 configurations, the authors claim that sparse pre-training can yield models with similar performance to dense pre-training, offering a more efficient approach by beginning with dense training and progressively sparsifying. However, due to limitations in hardware and software, the paper lacks direct evidence of computational savings, which is a primary motivation for sparse pre-training.

**Strengths:**

+ Modifying the Chinchilla scaling law to consider average active parameters is a novel approach, bridging dense and sparse pre-training frameworks effectively
+ The study evaluates a wide array of configurations and sparsity schedules, providing a thorough analysis of optimal sparse pre-training practices

**Weaknesses:**

- Despite the theoretical focus on efficiency, the paper lacks a demonstration of actual computational savings due to the current limitations in sparse matrix support in hardware/software, which could weaken the case for real-world applicability
- The study uses evaluation loss as the sole metric, without investigating the effects on real-world downstream tasks. This limits the ability to gauge the model’s practical effectiveness or generalization capabilities
- The framework’s reliance on finely tuned, phase-specific compute allocations (dense, pruning, recovery) introduces implementation complexity, which could be challenging to replicate or scale, particularly in resource-constrained environments

**Questions:**

See weakness

---

> ### Author Response · Authors · 2024-11-25
> **Author Response**
>
> We thank the reviewer for their constructive feedback.
>
> Here're our responses:
>
> > Despite the theoretical focus on efficiency, the paper lacks a demonstration of actual computational savings due to the current limitations in sparse matrix support in hardware/software, which could weaken the case for real-world applicability
>
> We acknowledge (as we did in the limitation section of our paper) that this work does not demonstrate actual computational savings. However, we believe such analysis is a critical and necessary step that informs whether investment in more sophisticated hardware/software support for sparsity is warranted.
> We also emphasize that our work has theoretical implications, too: Theoretically, it suggests that architectural changes due to sparsity do not change the rate of convergence of excess risk (sample complexity). In other words, for the data we model in practical scenarios, theorists should not expect to show that learning rate improves in terms of the samples needed with sparsity.
>
> > The study uses evaluation loss as the sole metric, without investigating the effects on real-world downstream tasks. This limits the ability to gauge the model’s practical effectiveness or generalization capabilities
>
> To assess the model's practical effectiveness and generalization capabilities, we replicated our claims in Fig.1 at **practical scale**, with **downstream task evaluations**. As a reminder, Fig.1 shows that with all other factors held constant, the average number of active parameters is an effective predictor of the final evaluation loss for sparse models, just as it is for dense models.
>
> To replicate at practical scale, we trained a sparse model with a practically relevant initial size of 1.14B parameters, pruned to 50% sparsity over 144B tokens, and a dense model with matching average number of active parameters during training. Both models were trained for the same number of tokens under an identical compute budget (6.8E20 FLOPs), approximately 5x the Chinchilla-optimal budget for a 1B-parameter model. This budget was chosen to complete training within the rebuttal period.
>
> We then evaluated both models on four commonly used NLP benchmarks to assess their downstream task performance.
> - **PIQA (Physical Interaction Question Answering)**: This dataset evaluates a model's ability to reason about everyday physical commonsense, such as choosing the most plausible way to perform a physical task.
> - **ARC (AI2 Reasoning Challenge)**: ARC consists of grade-school science questions designed to test reasoning and application of scientific knowledge, divided into "Easy" and "Challenge" subsets.
> - **Lambada**: This dataset tests a model's ability for long-range coherence by requiring it to predict the last word of a passage, where understanding the full context is critical.
> - **Winogrande**: This dataset evaluates commonsense reasoning by asking the model to resolve pronoun references in ambiguous sentences.
>
> | Model Name           | PIQA  | ARC Easy | ARC Challenge | Lambada | Winogrande | Geomean Accuracy |
> |----------------------|-------|----------|---------------|---------|------------|------------------|
> | 1.1B 50% Sparse      | 69.7  | 42.05    | 24.74         | 36.52   | 51.85      | 42.42            |
> | Matching Small Dense | 69.26 | 41.58    | 26.37         | 34.62   | 52.41      | 42.45            |
>
> We do not observe systemic differences in the sparse versus dense evaluation results; despite the matching dense model being 1.34x the size of the sparse model, they both achieve similar average accuracy (42.42% vs 42.45%). This demonstrates that, with all other factors held constant, the average number of active parameters is an effective predictor of **downstream task performance** for both sparse and dense models. We also confirmed that our original claim about evaluation perplexity holds in this case, with the sparse model achieving a perplexity of 2.44 and the matching small dense model achieving 2.46.

---

> > ### Author Response · Authors · 2024-11-25
> > **Author Response (Continued)**
> >
> > >The framework’s reliance on finely tuned, phase-specific compute allocations (dense, pruning, recovery) introduces implementation complexity, which could be challenging to replicate or scale, particularly in resource-constrained environments
> >
> > **Pruning phases.** The use of phased pruning schedules is well-established and not unique to our framework. Prior research, such as gradual pruning [1] and linear mode connectivity [2], has demonstrated the effectiveness and necessity of dividing the pruning process into distinct phases. These phases are crucial for accommodating the learning dynamics of sparse training [2] and for making optimal use of training compute, which is especially important given the high cost of training large language models. Our framework builds on this established practice by further refining phase-specific compute allocations to enhance the efficiency of sparse training.
> >
> > **Replication/scalability.** Importantly, our work explicitly addresses the issue of replication and scalability. Our experiments demonstrate the effectiveness of our sparse pre-training recipe across a range of model sizes varying by an order of magnitude. Our results showcase the robustness and adaptability of our approach, making them especially impactful in resource-constrained environments.
> >
> > [1] To prune, or not to prune: exploring the efficacy of pruning for model compression, arXiv 2017.
> >
> > [2] Linear Mode Connectivity and the Lottery Ticket Hypothesis, ICML 2020.

---

> ### Author Response · Authors · 2024-11-27
> **Follow-up**
>
> Dear reviewer tYDa:
>
> We are eager to hear whether our responses have sufficiently addressed your concerns. Please let us know if there are any remaining questions!

---

> > ### Author Response · Authors · 2024-12-02
> > **Follow-up**
> >
> > Dear reviewer tYDa:
> >
> > Hope to bring this to your attention again. We are eager to hear whether we have sufficiently addressed your concerns!
> >
> > Best wishes,
> > Authors of 9129.

---

> > > ### Comment · Reviewer_tYDa · 2024-12-02
> > >
> > > Thank the authors for addressing most of my concerns. I would like to raise my rating.

---

### Official Review · Reviewer_poHq · 2024-11-04

**Soundness:** 4
**Presentation:** 3
**Contribution:** 3
**Rating:** 8
**Confidence:** 4

**Summary:**

This paper conducts an overwhelming analysis of the effect of sparse pretraining on large language models (LLMs). Specifically, it proposes a new scaling law that is modified from the Chinchilla scaling law using a novel concept of the average number of active parameters during training, i.e., the averaged number of parameters that receive gradients in LLM pretraining among all training steps. The experimental results show that the proposed new scaling law sufficiently fits the evaluation loss prediction of LLM pretraining, while leveraging sparse pretraining could match the sparse, pruned model's evaluation loss to a larger, dense LLM.

**Strengths:**

- This paper extends the Chinchilla scaling law and adapts it to the scenario of sparse pretraining, where the modified scaling law uses the concept of averaged activate parameter number to sufficiently model the evaluation loss of LLMs.
- When design a pretraining pruning schedule using the scaling law proposed in this paper, one can pretrain an LLM with fewer parameters yet matching the evaluation performance of a dense model.
- The theoretical and empirical analysis in this paper demonstrates strong proof of the scaling law proposed in this paper.

**Weaknesses:**

- The scale of models being used in this paper is limited, as the authors have addressed in the limitation section. Larger model experiments are definitely useful and could bring wider impact to this paper.
- The only evaluation metric being used in this paper is the pretrained model's perplexity (or pretraining evaluation loss) without downstream task evaluations. As models could forget knowledge in pretraining because of pruning, a finer-grained analysis with task-specific evaluations could be beneficial.

**Questions:**

- What is the exact pruning method being used in the iterative pruning phase? A detailed description of the pruning criteria could be helpful for the audience to understand (e.g., structured/unstructured pruning and how is the pruning decision being made)
- $\bar{N}$ and $D$ are independent variables shown in equation (2), however, there seems no analysis in this paper showing that the effects of $\bar{N}$ and $D$ bring to the model's evaluation loss are independent.

---

> ### Author Response · Authors · 2024-11-25
> **Author Response**
>
> Thank you for your constructive feedback and your kind words!
>
> Here are our responses to questions and concerns:
>
> > The scale of models being used in this paper is limited, as the authors have addressed in the limitation section. Larger model experiments are definitely useful and could bring wider impact to this paper.
>
> While we cannot train larger models in the over-training regime explored in the paper (10x Chinchilla optimal and beyond) fast enough to finish within the rebuttal period, we have trained a pair of sparse/dense models with a smaller compute budget and matching average parameter count to demonstrate our finding in Fig.1 scales. As a reminder, Fig.1 shows that with all other factors held constant, the average number of active parameters is an effective predictor of the final evaluation loss for sparse models, just as it is for dense models.
>
> The sparse model starts from a 1.14B model and is pruned to 50% sparsity over 144B tokens and the dense model matches its average parameter count and trains for the same number of tokens. This compute budget (6.8E20 FLOPs) corresponds to ~5x Chinchilla optimal for the 1B model.
>
> With this pair of models, we confirmed that our Fig. 1 result scales: the average parameter count again is strongly predictive of model final evaluation loss. Specifically, the sparse model achieved a perplexity of 2.44, while the matching dense model achieved 2.46. Notably, at the end of pre-training, the matching dense model was 1.34x the size of the sparse model.
>
>
> >The only evaluation metric being used in this paper is the pretrained model's perplexity (or pretraining evaluation loss) without downstream task evaluations. As models could forget knowledge in pretraining because of pruning, a finer-grained analysis with task-specific revaluations could be beneficial.
>
> We re-evaluate our claims in Fig.1 using downstream task evaluations. We pick tasks that require varying combinations of knowledge recall and reasoning capabilities. We again show that, with other factors held constant, pairs of sparse and dense models trained with matching average parameter count yields similar downstream eval performance. Similar to Chinchilla[4], we focus our evaluation on the largest models trained, which is the 1.14B model with 50% sparsity and its smaller dense model counterpart from our previous answer.
> We used four downstream eval benchmarks:
> - **PIQA (Physical Interaction Question Answering)**: This dataset evaluates a model's ability to reason about everyday physical commonsense, such as choosing the most plausible way to perform a physical task.
> - **ARC (AI2 Reasoning Challenge)**: ARC consists of grade-school science questions designed to test reasoning and application of scientific knowledge, divided into "Easy" and "Challenge" subsets.
> - **Lambada**: This dataset tests a model's ability for long-range coherence by requiring it to predict the last word of a passage, where understanding the full context is critical.
> - **Winogrande**: This dataset evaluates commonsense reasoning by asking the model to resolve pronoun references in ambiguous sentences.
>
> | Model Name           | PIQA  | ARC Easy | ARC Challenge | Lambada | Winogrande | Geomean|
> |----------------------|-------|----------|---------------|---------|------------|------------------|
> | 1.1B 50% Sparse      | 69.7  | 42.05    | 24.74         | 36.52   | 51.85      | 42.42            |
> | Matching Small Dense | 69.26 | 41.58    | 26.37         | 34.62   | 52.41      | 42.45            |
>
> We do not observe systemic differences in the sparse versus dense evaluation results; despite the matching small dense model being 1.34x the size of the sparse model, they both achieve similar average accuracy (42.42% vs 42.45%); in other words, with all other factors held constant, the average number of active parameters is an effective predictor of the downstream task evaluation for sparse models, just as it is for dense models.

---

> > ### Author Response · Authors · 2024-11-25
> > **Author Response (Continued)**
> >
> > >What is the exact pruning method being used in the iterative pruning phase? A detailed description of the pruning criteria could be helpful for the audience to understand (e.g., structured/unstructured pruning and how is the pruning decision being made)
> >
> > In the iterative pruning phase, we employ global iterative magnitude pruning similar to [1][2][3]. At each step, a fixed fraction of prunable weights (from all linear layers) is removed. We rank the remaining weights by magnitude as a measure of importance and prune those with the smallest values globally.
> >
> > We have added additional clarifications to Section 3. Preliminary under Algorithm paragraph. We appreciate the reviewer for this helpful feedback.
> >
> > > N¯ and D are independent variables shown in equation (2), however, there seems no analysis in this paper showing that the effects of N¯ and D bring to the model's evaluation loss are independent.
> >
> > Indeed, the independence assumption was introduced by Chinchilla[4] and we adopted this assumption. The intuition, according to Section D.2 of the Chinchilla paper [4] was that loss may be decomposed to 3 independent components: (1). irreducible loss corresponding to the entropy of the language, (2). Loss due to the size of the hypothesis space, (3). Loss due to empirical risk minimization (rather than optimizing for true risk). [4] argued that term (2) depends only on parameter count and term (3) only on the dataset size.
> >
> > We believe a similar line of arguments extends to sparse pre-training. The gradual removal of weights over pre-training will affect loss only through its impact on the size of hypothesis space.
> >
> > [1] The Lottery Ticket Hypothesis: Finding Sparse, Trainable Neural Networks, ICLR 2019.
> >
> > [2] Comparing Rewinding and Fine-tuning in Neural Network Pruning, ICLR, 2020.
> >
> > [3] To prune, or not to prune: exploring the efficacy of pruning for model compression, arXiv 2017.
> >
> > [4] Training Compute-Optimal Large Language Models, NeurIPS 2022.

---

> ### Comment · Reviewer_poHq · 2024-11-28
>
> Thank you for the response. Most of my concerns have been addressed here. As a result, I will keep my original score.

---

### Author Response · Authors · 2024-12-04
**Rebuttal Summary**

As we reach the rebuttal deadline, we sincerely thank all reviewers for making such a productive and constructive discussion period possible!

We are happy to see positive recognition of our work. Reviewer poHq described our analysis as *"an overwhelming analysis of the effect of sparse pretraining on large language models (LLMs)."* Similarly, Reviewer Kf8Y highlighted that *"the analysis and empirical validation are solid and convincing. The findings are supported by both theoretical justifications and experiments over 80 sparse pretraining configurations."*

During the rebuttal period, we provided additional experimental validations to strengthen our claims:

- We extended our findings to models exceeding 1 billion parameters, demonstrating generalizability [[link](https://openreview.net/forum?id=ud8FtE1N4N&noteId=LfNH4MThhM)].
- We complemented our perplexity-based evaluations with downstream task results, covering benchmarks like PIQA, ARC, Lambada, and Winogrande [[link](https://openreview.net/forum?id=ud8FtE1N4N&noteId=LfNH4MThhM)].
- We conducted detailed analysis into how sparse scaling laws can guide practical LLM training, emphasizing their implications for optimizing both training and deployment configurations and resource consumption [[link](https://openreview.net/forum?id=ud8FtE1N4N&noteId=v7jkCLPM8v)].

Thank you again for your time and thoughtful input, which have contributed to refining and improving this work.

Authors of 9129.

---

### Meta-Review · Area_Chair_q9yh · 2024-12-20

**Metareview:**

The paper proposes a new scaling law that modifies the Chinchilla scaling law to use the average number of active parameters during training, rather than just the total parameter count. The authors claim that this modified scaling law can effectively model the evaluation loss for both sparsely and densely pre-trained LLMs.

Overall, the paper makes a solid contribution to the understanding of sparse pre-training for large language models. The novel scaling law, extensive empirical analysis, and practical insights are valuable. While the paper has some limitations in model scale and evaluation metrics, the reviewers generally find the work to be sound, well-presented, and a good contribution to the field. Therefore, I recommend accepting it.

**Additional Comments On Reviewer Discussion:**

During the rebuttal period, the authors provided additional experimental validations to strengthen their claims. They extended their findings to models exceeding 1 billion parameters, demonstrating the generalizability of their approach. The authors also complemented their perplexity-based evaluations with downstream task results, covering benchmarks like PIQA, ARC, Lambada, and Winogrande.  The authors also conducted detailed analysis on how sparse scaling laws can guide practical LLM training, emphasizing their implications for optimizing both training and deployment configurations as well as resource consumption.

---

### Decision · Program_Chairs · 2025-01-22

Accept (Poster)